# A new perspective on the evolution of "Kawara" roof tiles in Ryukyu: A multidisciplinary non-destructive analysis of roof tile transition at Shuri Castle, Ryukyu Islands, Japan

Hiroaki Aoyama[1,2☯*], Kaishi Yamagiwa[3,4,5☯*], Wataru Taira[1,2,6☯], Takeshi Kon[1]

**1** Research Planning Office, University of the Ryukyus, Nishihara, Okinawa, Japan, **2** Research Facility Center, University of the Ryukyus, Nishihara, Okinawa, Japan, **3** Research Fellow of the Japanese Society for the Promotion of Science, Chiyoda, Tokyo, Japan, **4** Research Institute for Islands and Sustainability, University of the Ryukyus, Nishihara, Okinawa, Japan, **5** Center for Strategic Research Project, University of the Ryukyus, Nishihara, Okinawa, Japan, **6** Center for Research Advancement and Collaboration, University of the Ryukyus, Nishihara, Okinawa, Japan

☯ These authors contributed equally to this work.
* aoyama@lab.u-ryukyu.ac.jp (HA); yamagiwa@lab.u-ryukyu.ac.jp (KY)

**Data Availability Statement:** All relevant data are within the paper and its Supporting Information files.

## Abstract

A unique historical architecture was created at Shuri Castle (*Shuri-jo*) in the Ryukyu Islands by its "Kawara" roof tiles. After the 13th and 14th centuries, Kawara tiles were introduced to the Ryukyu Islands from several regions, including China, Korea, and mainland Japan, and evolved shapes and patterns that are unique to this island region. However, the transition of some internal features, such as the chemical components and microstructure, had not been analyzed. This study used a multi-faceted approach for such internal data and non-destructive quantitative methods to propose a new perspective on the evolution of historical Ryukyuan Kawara. We analyzed two styles of Ryukyuan Kawara from the 13th to 15th centuries and found that the material processing and firing conditions of the two styles were very similar, even though it had been suggested that they had different origins. A quantitative analysis of tiles from the 16th to 19th centuries revealed a transition in color tone to red, leading to the modern traditional Ryukyuan tiles; traces of changes in firing conditions were also found along with this transition. Finally, the study revealed that the evolution of Ryukyuan Kawara consisted of changed factors, e.g. surface color, and unchanged factors, e.g. paste density. Previous archaeological studies mainly focused on changing external characteristics, such as form and pattern; however, our analysis showed that the internal features changed, while the elemental composition and paste density remained constant from the appearance of the roof tiles until the 19th century. We propose that this is related to different responses of individual factors to external stressors, such as the social context, which may be common to other archaeological artifacts as well. Our study provides a new perspective on the evolution of Ryukyuan Kawara and presents a different discussion of and methods for the chronological study of material culture.

**Funding:** This research was funded by a grant from the Shuri-jo Revival Research Project of the University of the Ryukyus. The funders had no role in study design, data collection and analysis, decision to publish, or preparation of the manuscript.

**Competing interests:** The authors have declared that no competing interests exist.

## Introduction

The Ryukyu Islands in the southwestern Japan archipelago have a unique history compared to surrounding regions, such as mainland Japan (e.g., Kyushu, Honshu), Taiwan, and China (Fig 1A). Shuri Castle (*Shuri-jo*) on Okinawa Island is representative of the architecture of the Ryukyu Islands and their culture and was used as a fort, castle, or center of rule mainly during the Ryukyu Kingdom period (AD 1429–1879). It is now registered as one of the World Heritage Sites of this region [1]. The architecture of Shuri Castle is characterized by "Aka-gawara" (means red roof tile), a material associated with traditional houses in Okinawa (Fig 1B). The Castle burned down multiple times, including in a tragic fire in 2019, and each time it was rebuilt using a large number of tiles. The ancient roof tiles at Shuri Castle therefore embody the history of tile culture in the Ryukyu Islands.

Kawara roof tiles in Japan are a building material designed mainly to keep out rain, sunlight, or wind and are generally known to be more durable and fire-resistant than shingle or

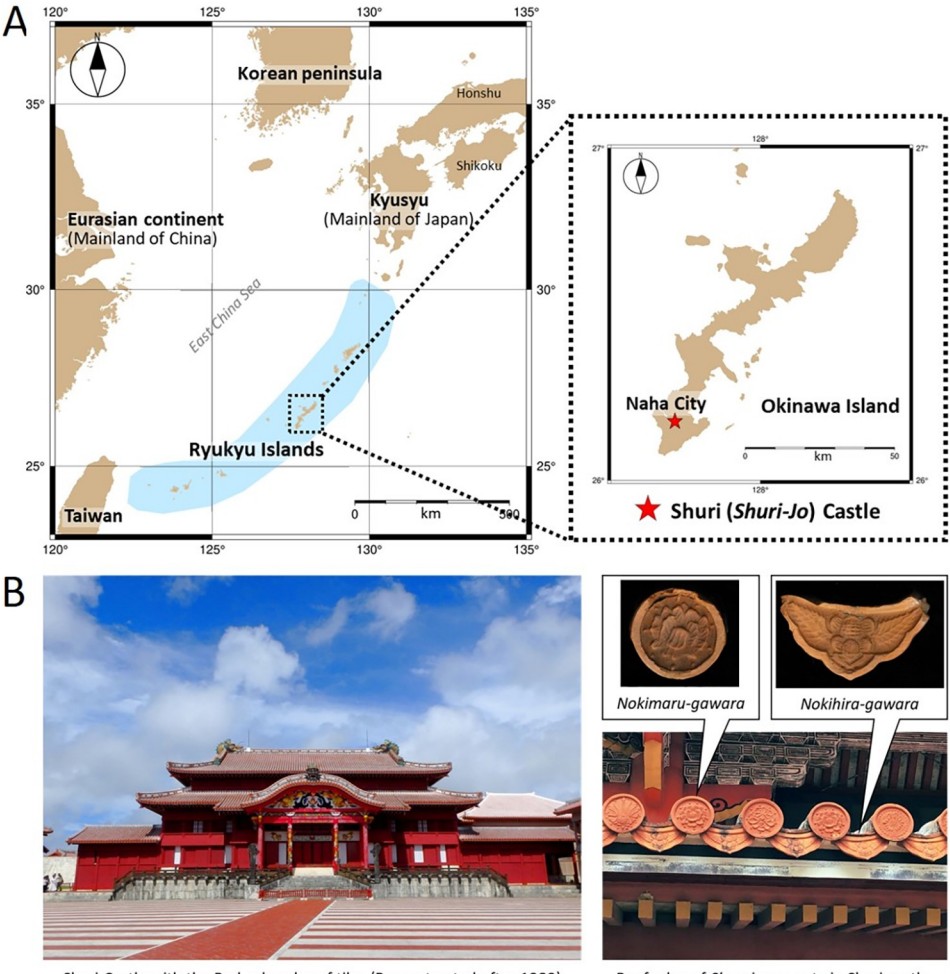

**Fig 1. Location of Shuri Castle and exterior view of Main Hall.** (A) Location of Shuri Castle (*Shuri-jo*) on Okinawa Island, the main island in the Ryukyu Islands, which are located between mainland Japan (Kyushu, Honshu, etc.), eastern Eurasia (southeastern China), and Taiwan. (B) Exterior view of the Main Hall and two typical types of roof tile: Semi-cylindrical "Maru-gawara" and flat "Hira-gawara" tiles. The Maru-gawara at the eaves are called "Nokimaru-gawara", and the Hira-gawara at the eaves are called "Nokihira-gawara" (or "Tekisui-gawara"). The various patterns decorating these eave-end tiles often serve as indicators of their age.

thatch roofing. In East Asia, ancient roof tiles made of ceramics were notably developed after the Warring States period in China (ca. fifth to third century BC) and were introduced into mainland Japan through the Korean peninsula until AD 538 (in the Kofun period) [2]. Initially, the roof tiles were mainly used for certain Buddhist temples, spread widely throughout various regions of mainland Japan [3], and eventually became a common roofing material for houses in Japan. In the Ryukyu Islands, Kawara was introduced after about the 13th or 14th century AD, when multiple types of roof tiles appeared with several origins assumed to be mainland Japan, the Korean peninsula, or mainland China. The ancient roof tiles, as a typical artifact, were evidence of the prosperous international relations between the Ryukyu Islands and its surrounding countries.

Ryukyuan ancient roof tiles have been broadly classified into three styles, "Kourai", "Yamato", and "Minchō", depending on their external characteristics, such as design and form. This classification is based on chronological order and the presumed origins of culture and technology (S1A Fig in S2 Appendix). The oldest Ryukyuan roof tiles appeared at least as early as the 13th to 14th centuries AD (This period corresponds to the Gusuku period in 10th to 14[th] century of Ryukyu Islands) and are presumed to have been used in Gusuku, the large stone-walled castles and fortresses that are representative of this period. The oldest tiles are classified into two styles: Kourai and Yamato. The Kourai style is associated with Korea's Goryeo Kingdom (AD 918–1392) due to external characteristics, such as design, shape, paddling traces, imprinting, and inscriptions on their surface [4–6]; specifically, they are assumed to be connected to refugees from the failure of the "Sambyeolcho" Rebellion in Goryeo (AD 1273) [7]. The appearance of the Yamato-style tile differs from that of the Kourai; based on external characteristics, such as design, form, and traces, it is presumed to have originated in Japan, possibly Kyushu, between the Kamakura (AD 1185–1333) and Muromachi (AD 1336–1572) periods. Although the two styles overlap chronologically, some scholars consider the Kourai to be older, as the Yamato incorporates techniques that resemble the Kourai traces [6,8,9].

In the 14th and 15th centuries AD, the Okinawa Island was unified into the Ryukyu Kingdom and became part of the Chinese dynasty's tributary system. A new style of roof tile (Minchō) appeared between the 16th and 17th centuries whose appearance was clearly different from previous styles, and its technical relationship with these earlier types is not well understood. Although the Ryukyu Kingdom was invaded and became a vassal state under the Japanese Satsuma feudal domain after AD 1609, some previous researchers have asserted that the new style of roof tile reflected cultural influences from the Ming dynasty, based on a comparison with roof tiles in the Chinese dynasty, for example, as described in the Chinese historical record "Tiangong Kaiwu" (published in AD 1637) [8,10].

However, there are many aspects of the Ryukyuan Kawara that have not been explained. These tiles, considered to have been introduced from surrounding regions, developed into unique tiles in the Ryukyu Islands, but the background of their uniqueness remains unclear. For example, one characteristic of Ryukyuan tiles is that they transitioned from gray to red without the use of glaze [11], but the reason for this is not known, although some hypotheses have been proposed. In addition, few changes other than in external characteristics have been analyzed. Most previous studies have focused on external features to show the chronology of Ryukyuan roof tiles, but limited consideration has been given to internal factors, such as raw material characteristics, paste condition, or internal structure. In the study of cultural evolution, various functional and stylistic factors influence artifacts, and the evolution of material culture cannot necessarily be discussed based on a single factor. Internal features are important factors when considering changes in the properties and functions of roof tiles and their production. However, as there is no effective method of analyzing tile characteristics, which are difficult to observe or quantify, quantitative or statistical analyses of these features are lacking.

For example, the color changes considered characteristic of Ryukyuan tiles have not been subjected to a quantitative analysis, and paste analysis through powder X-ray diffraction or petrographic observation has been limited to a few roof tiles due to the destruction of the materials; therefore, a statistical analysis was not possible.

In this study, we analyzed the internal features of the ancient Ryukyuan roof tiles at Shuri Castle quantitatively and statistically using non-destructive methods, an X-ray fluorescence microscope, a computed tomography (CT) scanner, and a digital image scanner. The results were then compared with the chronological transition of roof tiles based on their external characteristics, and finally, the process of roof tile transition in the Ryukyu Islands, its factors, and the mechanism of evolution were discussed.

## Materials and methods

### Site information

Shuri Castle is viewed as a special architectural resource due to the large number of tiles of various ages (including the three styles mentioned in this article) that have been excavated there. Although the Castle's origins remain unclear, some historical and archaeological research has suggested that the main palace was constructed in the mid-15th century ("Ankoku-zan Jukaboku no Kihi": inscription on stone monument built in 1427). It is presumed to have had a tile roof at this time because of the large number of tiles excavated [12–14]. This main palace was destroyed by fire at least three times (AD 1453, 1660, 1709) ("Kyuyou", a historical record written in AD 1743–1745). Records show that it was rebuilt with a single roof after the first fire in AD 1453 and that it was tiled when rebuilt after the second fire in AD 1670. [12,15–17] (S1B Fig in S2 Appendix) In addition, records of renovations for damage caused by storms, earthquakes, and deterioration exist, and it is presumed that tiles were produced and replaced each time [15,18]. Previous studies of ancient roof tiles have shown no significant difference in the tiles used at Shuri Castle and other sites [19]. Therefore, the changes in roof tiles identified at the Castle can be considered to generally reflect the historical transition of Ryukyuan roof tiles.

### Materials

As a sample, we used a total of 39 ancient and modern roof tiles from Shuri Castle (S1 Table in S1 Appendix) that comprised three ancient styles, including Kourai (KOR) (S2 Fig in S2 Appendix), Yamato (YMO) (S3 Fig in S2 Appendix), and Minchō (MCO) (S4 and S5 Figs in S2 Appendix), as well as modern roof tile samples (MDN) (S6 Fig in S2 Appendix). All the ancient samples were from the Ryukyu University Museum (Fujukan) and had been collected on the old Shuri campus of the University of the Ryukyus, which was on the site of Shuri Castle during 1950–1979. The modern samples were extracted from the debris of the Main Hall, which burned down during the Shuri Castle fire on October 31, 2019; these were associated with the recent reconstruction of Shuri Castle after 1989 and made by modern roof tile manufacturing technology.

Although the ancient roof tiles in our sample were not scientifically chronologically dated (e.g., using radiocarbon dating techniques), they can be classified according to their style and dated based on their external characteristics. We classified all the ancient roof tiles into three basic types: KOR, YMO, and MCO. In addition, based on the chronological classification established by Uehara [6], we separated MCO into two phases: Period I (early 17th century) and Period II, III, and IV (late 17th to mid-19th century). Some scholars have argued that MCO may have appeared in the 16th century [20]; however, only tile types from the early 17th century onward have been definitively established.

## Image scanning and color distance analysis

Roof tile images were collected using the scan function of the bizhub C368 multifunction printer (Konica Minolta, Japan). All images were scanned under the same conditions as the Kodak color control patches (Kodak, USA). The scanned images were trimmed to a 1 cm-square region without shadows or patterns. The average color information in two models, RGB and Lab, was quantified from the trimmed images using GIMP 2.10.20 software (http://www.gimp.org/). Traditional reddish colors were used from the DIC Color Guide's "Traditional Colors of Nippon" series (DIC Graphics Corporation, Tokyo, Japan).

Color distance was measured according to CIEDE2000, which takes into account human perception, as suggested by the International Commission on Illumination [21]. Color distances were calculated using the delta_e_cie2000 function in the Python module (python-colormath) [22].

The color distance data were clustered using the hclust function of the stat package in R. The heatmap was drawn using the heatmap.2 function of the gplots package in R. A multidimensional scaling (MDS) analysis was performed using the cmdscale function of the stat package in R and drawn using the dpyr, tidyverse, and ggpubr packages in R. Box and jitter plots were performed using Past software (ver. 4.03). The Steel-Dwass test was performed using the psDCFlig function of the NSM3 package in R.

## X-ray fluorescence microscopic analysis

The analysis in this study improved on our previous study [23,24]. An XGT-7200 XRF microscope (Horiba, Japan) was used to analyze the chemical components of the roof tiles, enabling measurement of the elemental distribution on the tile surfaces using spectral analysis. X-rays were generated at 30 kV/1 mA; further, the mass ratio of elements on a measuring point was calculated using 10-μm irradiation diameters over a preset time of 60 s. We performed this analysis with 231 multipoints set at intervals of 1 mm in 20 mm × 10 mm squares for the surface of each roof tile. The X-ray sources were based on rhodium (Rh); each chemical composition was calculated as an oxide based on fundamental parameter methods, and all the elemental ratios of the spectral analysis were described in terms of percentage data for mass.

The average values were calculated based on the Dirichlet distribution using 231 measuring points, as described in our previous study [24]. A principal component analysis (PCA) was performed using the prcomp function of the stat package in R. Linear discriminant analysis effect size (LEfSe) was performed using the tool within the Galaxy web application [25]. A p-value of $< 0.05$ was considered significant in both Kruskal–Wallis and pairwise Wilcoxon tests, and a score of $\geq 2.0$ was considered the threshold on the logarithmic linear discriminant analysis (LDA) score for discriminative features.

## X-ray computed tomography (CT)

X-ray CT images of the tile were acquired using a Nikon XT H225 (Nikon, Tokyo, Japan). As the dimensions and shapes differed greatly for each sample, the radiographic conditions could not be standardized. The radiographic conditions for voltage, current, exposure time, and the use of metal filters were changed while maintaining the brightness distribution of each image within a range of 10,000–60,000. A few samples did not meet the brightness requirements due to shape issues. In those cases, we checked the noise in the 3D reconstructed image and used the data if there appeared to be no major problems. The process for the difference in resolution between samples is described in the supplementary methods in S2 Appendix.

## Pores identification, counting, and measurement

CT Pro 5.4 software (Nikon, Tokyo, Japan) was used to perform 3D image reconstruction and image correction (i.e., beam hardening and noise reduction) was used as needed. All pores were identified, counted, and measured, according to standard protocols, by the Porosity/ Inclusion Analysis Module in VG Studio MAX 3.4 (Volume Graphics GmbH, Heidelberg, Germany), but no thresholding by defect probability values were used in this process. Using this software, we obtained data on the number, volume, and character values of pores (i.e., defect probability value, compactness, sphericity) and volume of material paste. The software package NYSOL version 2.4.2 (NYSOL Corporation, Osaka, Japan) [26], was used to perform data wrangling, such as unification of detection limits and thresholding defect probability values (see Supplementary Materials and Methods in S2 Appendix).

## Eight values for internal microstructure features

The following eight values were calculated to represent the internal microstructural features of each sample: (1) Density-S: weight density of the sample ($g/cm^3$); (2) Density-M: weight density excluding pores (i.e., weight density of material paste); (3) Median-vol: median pore volume ($g/cm^3$); (4) Mode-vol: most frequent pore volume ($g/cm^3$); (5) Dispersion-vol: dispersion of pore volume; (6) RLV: ratio of large pores to total pore volume; (7) Num-density: pore number density ($n/cm^3$); and (8) RLN: ratio of large pores to total number of pores. For details of each value, see the supplementary methods in S2 Appendix.

## Consideration of data reliability

We checked whether pore data acquisition and data wrangling were successful. First, violin plots were drawn to determine the distribution of pore volumes for each sample (S7 Fig in S2 Appendix). In MCO020, the distribution (especially the lowest value) appeared to be very different from that of the other samples. This suggested the possibility of outlier generation during CT imaging and data filtering. Therefore, we performed outlier tests ($Q$ test) for Median-vol for each style using Dixon's method. These were performed in R with the dixon.test function in the {outliers} package. MCO020 (Median-vol: 0.71 $mm^3$) was detected as an outlier in the Minchō style ($Q = 0.94$, p $<$ 2.2E-16), but there were no significant outliers in other styles. In the case of MCO020, its physical features (it was very large and had a complex shape) and analytical factors, such as the CT imaging resolution, resulted in abnormal data. Therefore, we removed the MCO020 data in subsequent analyses.

## Data analysis

Spearman's correlation analysis was performed using the cor.test function and stats package in R. The Steel Dwass test using the Monte Carlo method (with 10,000 iterations) was performed using the pSDCFlig function and the NSM3 package in R. The Wilcoxon rank sum test was performed using the wilcox.test function and stats package in R. The exact Wilcoxon rank sum test was performed using the wilcox.exact function and exactRankTest package in R. Nonmetric multidimensional scaling (NMDS) was performed using the metaMDS function of the vegan package in R.

## Results

### Quantitative color differences analysis

To analyze the color differences in roof tiles, we quantified the color based on the information on the roof tile surface acquired by the image scanner (S2 Table in S1 Appendix). Hence, we

measured the color distance between each roof tile using the CIEDE2000 method, which reflects differences in color perception (S3 Table in S1 Appendix). The result of clustering based on the color distance matrix showed two large reddish and grayish clusters, each containing two or three subclusters (Fig 2A). All the ancient roof tile styles were dispersed across multiple subclusters, especially the Minchō style, which was included within all subclusters

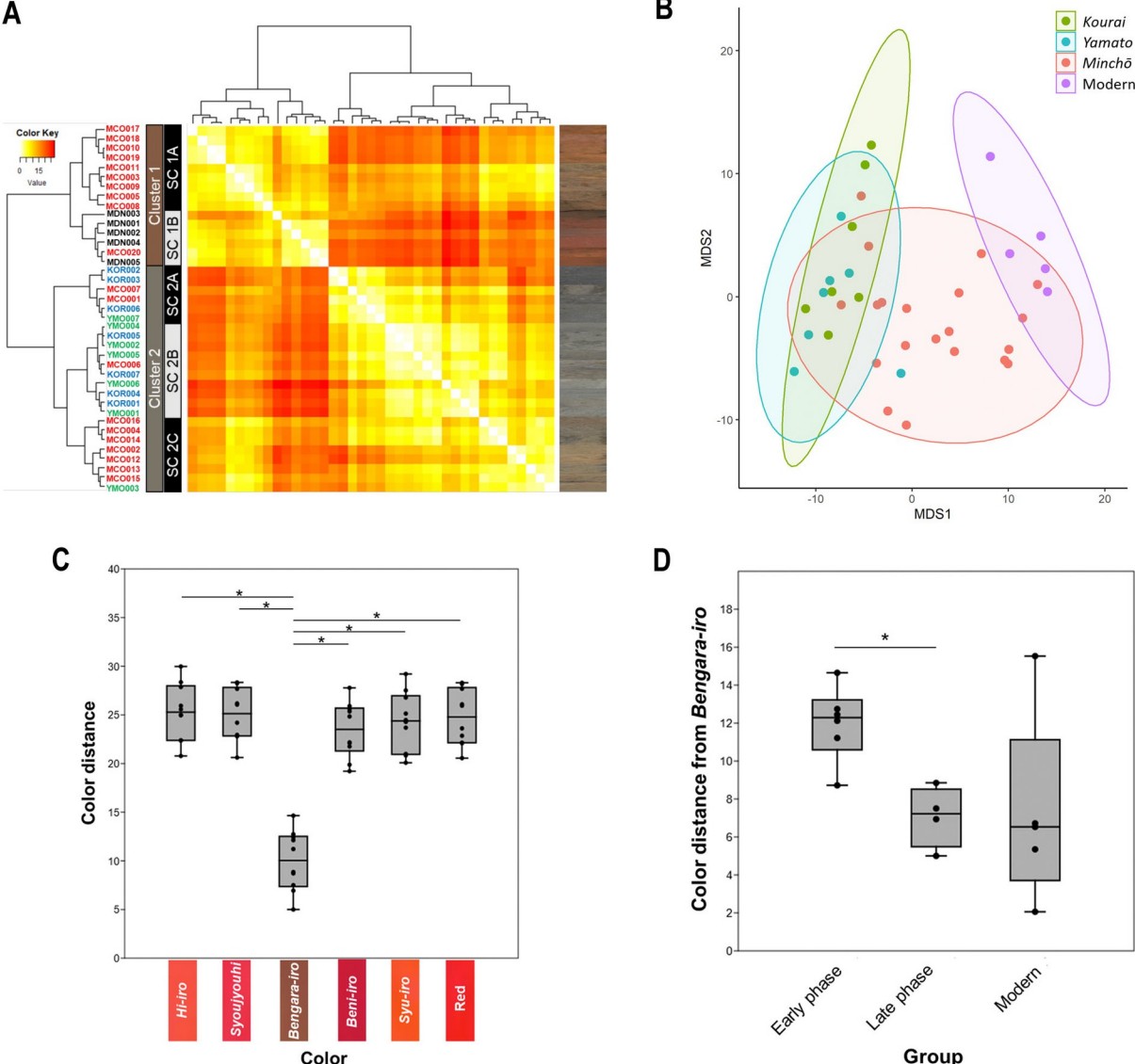

**Fig 2. Color distance analysis of roof tiles.** (A) Heatmap of roof tiles based on color distance. A light color indicates the color proximity based on the color distance of each roof tile. The left column shows clusters by dendrogram. Three ancient and one modern roof tile were classified into two clusters, including five subclusters: SC. The right column represents scanned images of each roof tile. MCO: Minchō style, KOR: Kourai style, YMO: Yamato style, MDN: Modern. (B) MDS of the roof tiles based on the color distance matrix: Classical MDS plot derived from the color distance matrix of each roof tile. The four polygons represent the convex hull of each roof tile style. (C) Box plot of color distance from six reddish colors. The points indicate the color distance between each color and the roof tiles that were classified in the red color cluster in Minchō style. The six reddish colors are five traditional Japanese reddish colors and a standard red color. "Bengara-iro" had a significantly smaller color distance from the roof tile color, compared with the other five colors (p < 0.001: Steel-Dwass test). (D) Box plot of color distance from Bengara-iro in both Minchō-style phases. The points indicate the color distance between Bengara-iro and the roof tiles in three groups: Early and late phases of Minchō style and the modern tile. The late Minchō phase had a significantly smaller color distance from Bengara-iro, compared with the early phase (p < 0.05: Steel-Dwass test).

and concentrated in two subclusters (SC 1A and SC 2C). In contrast, the Yamato and Kourai styles were only included in the grayish cluster, and both styles were distributed in multiple subclusters (SC 2A, SC 2B, and SC 2C). The modern tiles formed a single subcluster (SC 1B). We showed the relative distance to each roof tile using MDS analysis based on color distance (Fig 2B). The Minchō was widely distributed, compared with the other styles, and some of the tiles overlapped with other roof tile styles. These results suggest that Minchō had greater color diversity than other styles, but the grayish color of Kourai and Yamato differed slightly from that of Minchō.

Next, we focused on the reddish color of some Minchō roof tiles that are usually called Aka-gawara. There are multiple traditional reddish colors in Japan. Therefore, we analyzed the color distance between the roof tiles of the reddish cluster (Cluster 1) and each reddish color for five traditional Japanese reddish colors ("Hi-iro", "Shoujyouhi", Bengara-iro, "Beni-iro", and "Syu-iro") and a standard red color (Kodak color control patches) using the CIEDE2000 method (S4 Table in S1 Appendix). The results indicated that the reddish color of roof tiles was significantly similar to Bengara-iro significantly in these colors (Fig 2C). Furthermore, the color distance between the Bengara-iro and Minchō roof tiles was significantly less in the late phase, compared with the early phase (Fig 2D). These results suggest that the roof tiles in Cluster 1 was significantly similar to Bengara-iro, not the standard red color, and that the roof tile color became redder in the late phase of the Minchō style.

## Comparative analysis of chemical components

We measured the ratio of 11 elemental components of the roof tiles using a non-destructive method (S5 Table in S1 Appendix). According to the results of the PCA, each style of roof tile was not clearly separated from the others, but there were certain trends in their components that can be explained by three biplot vectors, $SiO_2$, $Al_2O_3$, and $Fe_2O_3$ (Fig 3A). Furthermore, the results of the nonmetric multidimensional scaling (NMDS) analysis indicated that the characteristics of elemental compositions in the ancient roof tiles overlapped with each other but not with the modern tile (Fig 3B). Moreover, the two Minchō-style phases overlapped (Fig 3C). Additionally, to detect the feature elements that explain the differences between the ancient roof tile styles in each elemental composition, we performed a linear discriminant analysis effect size (LEfSe). The results indicated that the Minchō style was featured by two elements, $Fe_2O_3$ and $K_2O$, and the Kourai style was characterized by $SiO_2$ within each style (Fig 3D). Furthermore, the comparison among the Minchō-style tiles indicated that the early phase (Period I) was featured by two elements, $Al_2O_3$ and $Fe_2O_3$, while the late phase (Period II and III) was characterized by SrO (Fig 3E). These results suggest that the ancient roof tiles were not clearly formed into independent clusters based on the elemental compositions. In contrast, the differences in roof tiles between styles or phases could be explained by specific elements in the roof tile paste.

## Physical feature analysis for the internal microstructure of the roof tiles

We measured the internal structures of the roof tiles non-destructively using a CT scan and obtained eight values indicating the features (S6 Table in S1 Appendix, S8 Fig in S2 Appendix). An MDS analysis was performed using these values to check the similarity of the internal microstructure among styles (Fig 4A). The MDS plots overlapped among the three ancient roof tile styles (Korai, Yamato, and Minchō), indicating that the internal microstructure was similar. In contrast, the modern roof tiles had a different internal microstructure from the other styles, with the center of its 95% probability ellipse being away from the distributions of the other styles.

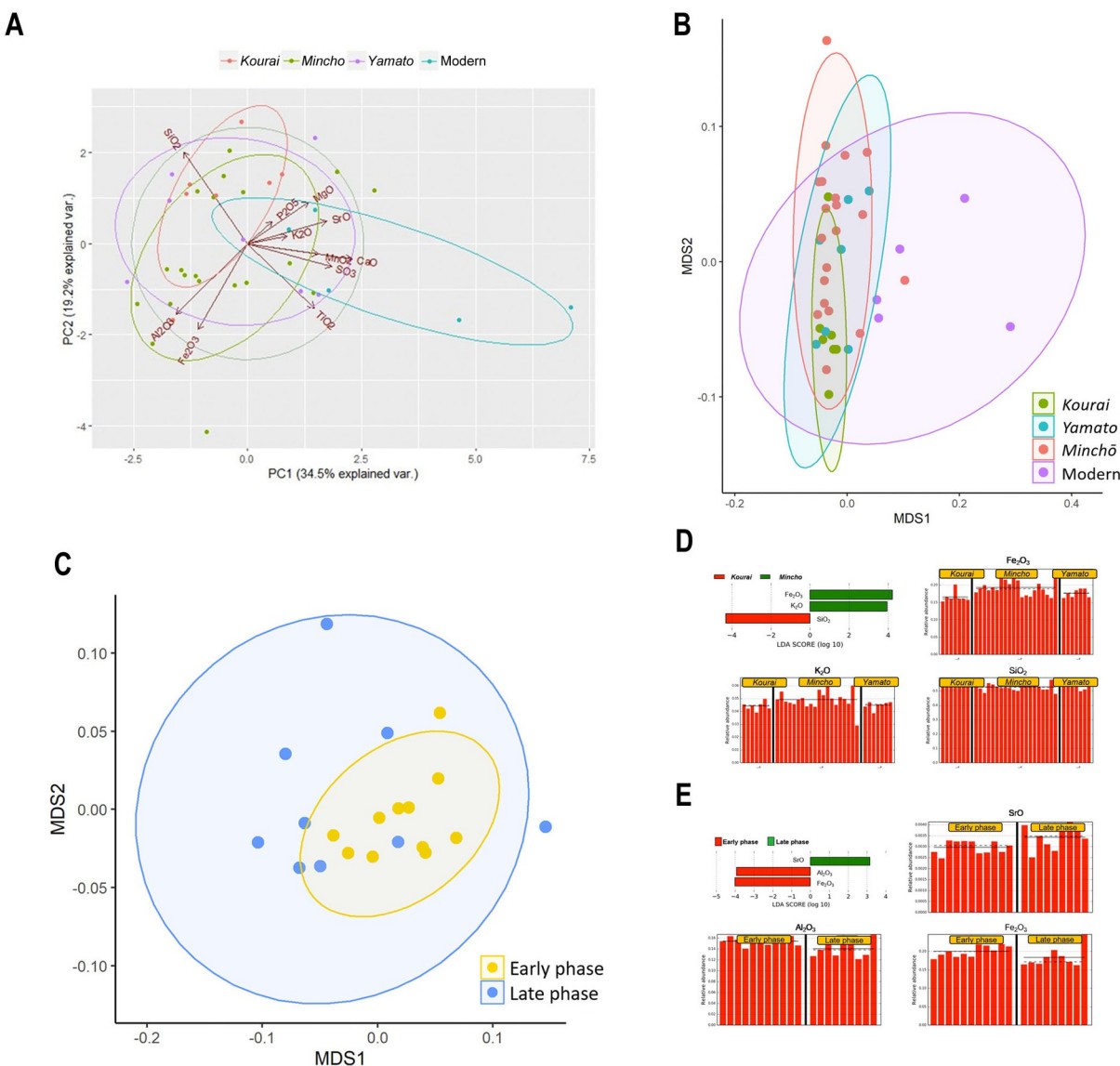

**Fig 3. Chemical component analysis of roof tiles.** (A) PCA of the roof tiles based on chemical components. The roof tiles were plotted by PCA to show the contribution rate of each element among the roof tiles. The ellipses were 95% confidence ellipses assuming a multivariate t-distribution. (B) NMDS of the roof tiles based on chemical components. The NMDS plot was derived from the chemical components of each roof tile. The ellipses were 95% confidence ellipses assuming a multivariate t-distribution. (C) NMDS of the Minchō-style roof tiles based on chemical components. The ellipses were 95% confidence ellipses assuming a multivariate t-distribution. (D) LEfSe of the chemical components in three ancient roof tile styles. The LEfSe revealed a list of features that enabled discrimination between the three ancient roof tile styles. The horizonal bar graph shows the LDA scores for the Kourai and Minchō styles. The histograms show the presence of three featured elements in the ancient roof tiles. The horizontal straight lines in the panels indicate the group means, and the dotted lines indicate the group medians. (E) LEfSe of the chemical components in two Minchō-style phases. The LEfSe revealed a list of features that enabled discrimination between the three ancient roof tile styles. The horizonal bar graph shows the LDA scores for the early and late phases. The histograms show the presence of three featured elements in the Minchō-style roof tiles. The horizontal straight lines in the panels indicate the group means, and the dotted lines indicate the group medians.

We then performed Kruskal-Wallis tests to check whether there were significant differences among the styles for each of the eight values (Fig 4B). Significant differences were found in seven values, the exception being Num-density. Next, post hoc tests were performed using the Steel-Dwass multiple comparison procedure to check which style combinations showed significant differences (S7 Table in S1 Appendix). As with the MDS results, most of the significant differences

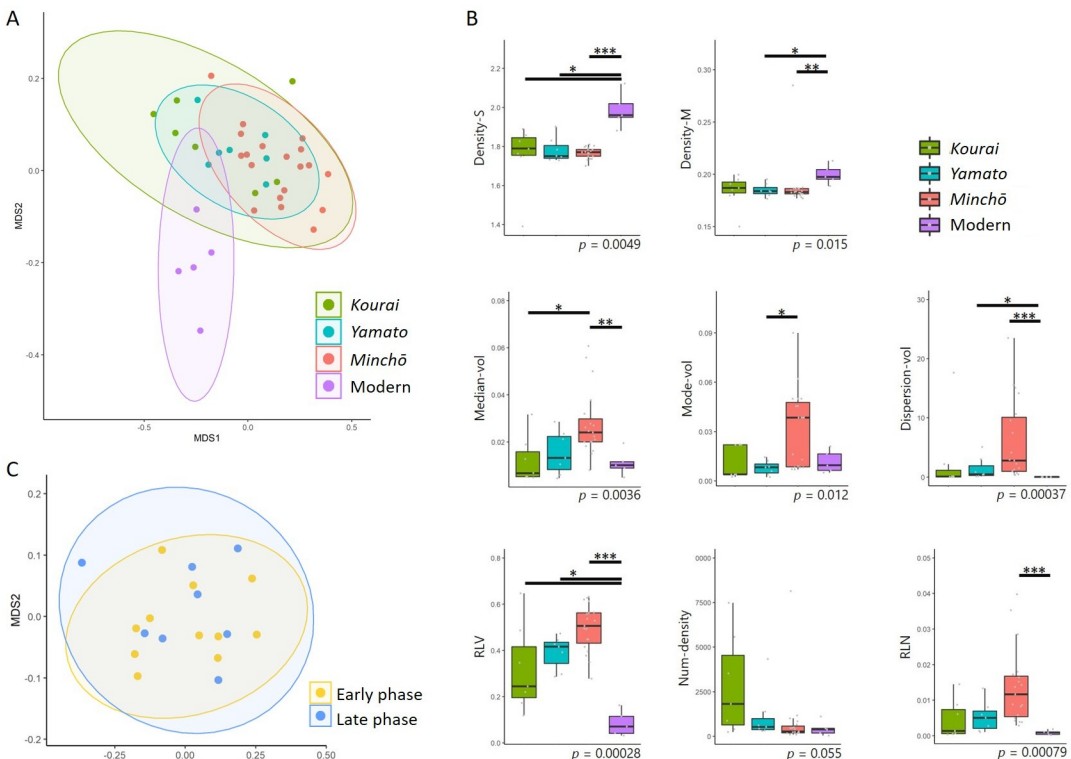

**Fig 4. Analysis of microstructural features.** (A) NMDS plots of three ancient and one modern roof tile style based on microstructural features. The NMDS plots were derived from the eight microstructural values for each tile. The ellipses were 95% confidence ellipses assuming a multivariate t-distribution. (B) Box plots comparing the three styles. The p-values in the Kruskal-Wallis tests are shown. Asterisks indicate statistically significant p-values in the Steel-Dwass test. *: p < 0.05, **: p < 0.01, ***: p < 0.001. The Steel-Dwass test was not performed for Num-density. (C) NMDS plots of Minchō-style roof tiles based on the microstructural features (eight values). The yellow plots and ellipse indicate the early phase (Period I), and blue indicates the late phase (Periods II and III). The ellipses were 95% confidence ellipses assuming a multivariate t-distribution.

were found between the modern tiles and the other styles. All values except Mode-vol were significantly different between the modern tiles and the Minchō style, while there were significant differences between the modern tiles and the Yamato style for four values and for two values for the Korai style. On Density-S, the modern tile was significantly different from all the other styles.

When focusing on the three ancient styles, there were significant differences between Minchō and Korai for Median-vol and between Minchō and Yamato for Mode-vol. This means that the Minchō tiles tended to have larger pores than the Korai and Yamato tiles. Contrastingly, there was no significant difference in Density-M among the three.

In addition, to investigate the differences between the Minchō-style phases, we compared the early phase (Period I) with the late phase (Periods II and III). An MDS analysis was performed using eight values, which showed that the distribution of points completely overlapped between the early and late phases, and the internal microstructure was similar (Fig 4C). Exact Wilcoxon rank sum tests were also performed, but no significant differences were found for any of the values (S8 Table in S1 Appendix).

## Correlation analysis of the factors contributing to the color change of the ancient roof tiles

To clarify the chemical components that affected the color change to reddish in Minchō tiles, we analyzed the correlation between the quantities of elements and the color distance from

Bengara-iro in each roof tile (S9 Table in S1 Appendix, S9A Fig in S2 Appendix). The results indicated a significant correlation between the color distance and four elements: $SiO_2$, $SO_3$, $K_2O$, and SrO. In particular, the amount of $SO_3$ showed the most significant negative correlation (rho = −0.59, p = 0.0058). Additionally, we performed an LEfSe analysis to determine the elemental features to explain the differences between the reddish and grayish color clusters in the Minchō-style tiles (S10 Fig in S2 Appendix). The results indicated that the reddish cluster was characterized by $SO_3$, and the grayish cluster was featured by $SiO_2$. This suggests that the change in color to reddish in the Minchō tiles was the result of an increase in sulfur content.

Next, to clarify the microstructural properties related to the change in color to reddish in the Minchō tiles, we analyzed the correlation between the properties of the microstructure and the color distance from Bengara-iro in each roof tile. The results showed that five properties, Mode-vol, Median-vol, Num-density, RLV, and RLN, were significantly correlated with the color distance from Bengara-iro (S10 Table in S1 Appendix, S9B Fig in S2 Appendix). We also analyzed whether there were differences in the properties between the two Minchō reddish and grayish color clusters. The results indicated that the pore volume mode was significantly different (S11 Table in S1 Appendix, p = 0.032). This result suggests that the roof tile pores became larger and more numerous as the tile color moved closer to Bengara-iro, with the pore size being especially strongly related to the change.

## Discussion

### 1. Transition from Kourai and Yamato to Early Minchō style

For the Kourai and Yamato roof tiles, which coexisted in the 14th to 15th centuries, there were external differences (shape, surface pattern, description, and imprinting) between the two styles, and it has been assumed that these differences are due to their different cultural origins and technological channels [6,9,20]. In contrast with these assumptions, however, our analysis did not detect any significant differences between the two styles in color tone, paste material characteristics, or internal structure (Figs 2B, 3B and 4A). In particular, there was no significant difference in paste characteristics between the two styles based on the overall elemental components and internal structures, suggesting that there was no significant difference in the processing of materials, firing conditions, or product quality of the two styles.

Roof tiles are not usually made from natural clay: the clay is processed by removing foreign bodies (e.g., large stones) and adding tempers to prepare the paste. Because there were no obvious foreign substances on the fractured surfaces of any of the samples, the tiles were thought to be refined. In the Urasoe Castle ruins, which are located on Okinawa Island and are older than those of Shuri Castle (ca. 13th to 15th century AD), there was no significant difference in the elemental composition of the Kourai- and Yamato-style tiles [27], and this finding was confirmed at Shuri Castle.

Because no clear traces of firing or production facilities for 13th- to 15th-century roof tiles have been found in the Ryukyu Islands, it has not been possible to conclude whether they were produced in this region or were imported. However, our results suggest that, at the very least, the technical background of the production of these two roof tile styles at Shuri Castle (perhaps also at Urasoe Castle) had similarities that can be detected in their color, paste characteristics, and internal structure. This might mean that both the Kourai and Yamato roof tiles were made in the Ryukyus.

The Minchō style is estimated to have been established around the 16th century AD [28], but the cultural or technological connections with previous styles, such as Kourai and Yamato, have not been clarified. Although some preceding studies have shown that the external characteristics of Minchō tiles are significantly different compared to previous styles [6,28], our results confirmed changes in Minchō color, raw materials, and firing temperatures.

First, our results for the gray tiles show that Minchō tiles have a different coloration from Kourai and Yamato tiles. Previous studies [6,29] had considered the gray tiles of all styles to be based on the same reduction flame, but the analysis of the color tone indicated variations in the reduction flame firing.

Second, significant differences in the elemental characteristics of the paste, such as $Fe_2O_3$, $K_2O$, and $SiO_2$, were found, mainly between the Kourai and Minchō styles. In contrast, the overall elemental components of Minchō and previous styles were similar; therefore, it is unlikely that the processing method, such as refining the material or adding admixtures, changed significantly. The differences in the characteristics of the paste found between the Kourai and Minchō styles may reflect differences in the origin or composition of the raw clay.

Third, we detected an increase in the size of the internal pores from previous styles to the Minchō (Fig 4B). It has been reported that the average pore radius increases with higher firing temperatures up to approximately 1000–1050°C [30], suggesting that the Minchō used a higher firing temperature.

However, the elemental composition of the paste and the density of the tiles did not change significantly from previous styles to the Minchō (Figs 3A, 3B and 4B), suggesting marginal change in the processing method for the paste and the quality of the tiles. There are two possible explanations for this finding. The first is that the soil conditions and product quality required for tile production were the same, regardless of the cultural or technological background. The second is that the commonality of material processing and product quality indicates technical continuity from the Kourai and Yamato to the Minchō. Our results suggest a new possibility—that the Minchō was influenced by Chinese patterns and forms but inherited techniques related to material processing and tile quality from the Kourai and Yamato. Although this possibility needs to be tested through comparisons with tiles from surrounding areas, it is a new possibility that emerged only by looking at tiles from a complex data perspective.

## 2. Unique color transition after Minchō style and its background

The Minchō-style roof tiles that appeared from the 16th to the 17th centuries had many chronological variations and were recognized as having developed their uniqueness in the Ryukyu Islands. In particular, the changes in patterns, shapes, and colors were considered characteristic. Of these, the change in color, from grayish to reddish, was recognized as a major turning point leading to the modern red roof tile (Aka-gawara), and previous studies have attributed this to changing firing conditions from reduction to oxidation [6,29,31]. However, it is thought that the changes did not all occur at once, because when temporally classified, the changes in the various roof tile patterns overlap [12]. Our results support this explanation of the color transition. In contrast, the quantitative analysis revealed that the color in the later phase, which was recognized as a single color in Minchō, resembled Bengara-iro more closely (Fig 2D). This also suggests that the change to Bengara-iro was gradual.

The proximity to Bengara-iro was positively correlated with the amount of sulfur (S9 Table in S1 Appendix and S10 Fig in S2 Appendix). Because the sulfur content in the two Minchō-style phases was not significantly different, the amount of sulfur in the paste correlated with the color, not the pattern. It has been reported that sulfur vapor at high temperatures in oxidation firing conditions reacts with hematite on the pottery surface to generate ferrous sulfide (FeS) [32]. In our case, sulfur that was indirectly present under the firing conditions vaporized and most likely sulfurized some of the iron on the roof tile surface during the oxidation firing. Firewood may contain trace amounts of sulfur, depending on species and growth environment [33] and is considered a candidate source of sulfur during burning. In addition,

the firing temperature may have increased because the pore size of the reddish cluster of roof tiles was significantly larger than that of the grayish cluster (S11 Table in S1 Appendix). Therefore, there are three possibilities for the progression of color tone toward Bengara-iro: 1) the oxidation of iron increased in the firing conditions with the rise in temperature; 2) the type of wood used and/or the amount of sulfur it contained changed; or 3) a combination of both. The detailed mechanisms will require additional research in the future. Interestingly, the historical and archeological data strongly suggest that the transition related to colors and elements was caused by changes in firing conditions and firewood. For example, the political consolidation of the ceramic production areas in Tsuboya (Naha, Okinawa) occurred in AD 1682 [6] and resulted in a change in kiln type from flat kilns to climbing kilns. In addition, there was an increase in demand for timber during the late 17th and early 18th centuries because of an increase in public works projects, such as the construction of the royal palace at Shuri Castle; industrial factors (i.e., shipbuilding and sugar production); and the firewood needs associated with population growth [34]. Limited firewood resources are considered to have contributed to the conversion to oxidative firing [6], and it is possible that the firewood itself changed.

The Bengara-iro colored Minchō-style roof tiles endured in the Ryukyu Islands until the 19th century. During this period, black smoked roof tiles named "Ibushi-gawara" were produced for use in symbolic castles in Japan (e.g., Edo Castle) [3], and glazed roof tiles, in colors such as yellow and green, were produced in China (e.g., Forbidden City) [35]. Although Ryukyu had the opportunity to import techniques from Japan and China, it did not use these techniques but developed its own unglazed reddish roof tiles. The reasons for this remain unclear, but as Ryukyu was in the complicated situation of being under the control of Satsuma while having a feudal relationship with China [36], it is possible that they did not prioritize the incorporation of a single country's technology. This suggests that the background to the appearance of colored roof tiles in Ryukyu encompasses not only the production technology and fuel resources, but also the socio-political background.

## 3. New perspective on the evolution of Kawara in Ryukyu

In this study, we analyzed the roof tiles from Shuri Castle and detected differences and similarities in color, material characteristics, and internal structure between past styles. Based on changes in each factor, we classified them into three transition patterns (Fig 5). Transition pattern 1 based on the external characteristics comprised the most variables. Transition pattern 2 consisted of three factors (color tone, elemental characteristics of the paste, and pore size), which had similarities in the Kourai and Yamato styles and differed in specific phases of the Minchō style. And transition pattern 3 consisted of two factors (whole chemical composition and paste density), which had similarities among all ancient roof tile styles. Transition patterns 2 and 3, newly revealed by our analysis, differed from the existing tile transition (transition pattern 1).

We suggest two hypotheses to explain the detection of the three transition patterns the roof tiles as material culture (Fig 6). The conventional hypothesis based on previous studies is that each roof tile trait was subjected to individual stress, which resulted in the formation of multiple transition patterns (Fig 6A). This resembles the previous view of cultural evolution. For example, Dunnel suggested that an artifact has multiple traits classified into style and function and supposed different evolution mechanisms between them [37]. Boyd and Richerson [38], also focused on separate evolutionary processes of multiple traits of material culture and considered that evolution results from stress that encourages the selection of various individual traits. However, in these explanations of cultural evolution, external stress seems to affect each

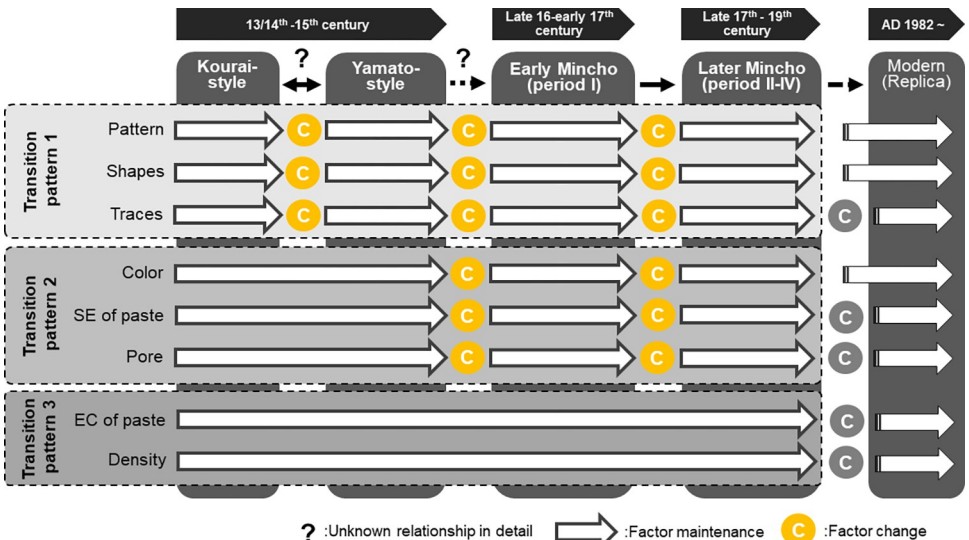

**Fig 5. Summary of roof tile evolutionary process.** Differences in appearance factors, such as patterns, forms, and traces between tile styles, from previous studies are shown (transition pattern 1). These differences are the basis for differentiating the roof tile style or period. Our analysis showed no significant difference between the Kourai and Yamato for color, specific elements (SE) of paste, and pores in paste; in contrast, after the emergence of the Minchō style, these factors were detected to have changed, along with the appearance characteristics (transition pattern 2). No significant difference in elemental composition (EC) or paste density were found between any of the ancient tile styles, only between them and the modern reconstructed tiles (transition pattern 3). Contrastingly, in the modern reconstructed tiles, the external factors did not change because they were designed as imitations, while internal factors such as SE, EC, pore, and paste density were confirmed to be significantly different.

trait individually, yet it is difficult to envision such a situation in practice. For example, if we proposed that the evolution of transition pattern 2 was caused by social stressors such as fuel shortages, such stressors must not only have affected the factors related to transition pattern 2 (color tone, materials, internal structure), but also the whole of the roof tile culture.

Therefore, we propose another possibility based on the conventional hypothesis that an external social or environmental stress affected the "whole" of the roof tile culture but that only limited traits reacted to it, resulting in the formation of various transition patterns (Fig 6B). Although it is not different from the previous view of cultural evolution in that each factor evolves differently, it shows a new correspondence between traits and external stresses. In the case of transition pattern 2, it is considered that the entire roof tile culture was subjected to the social stress caused by fuel shortages, but some of the firing methods showed an especially strong reaction to it, which led to a transition in the color and pore size of the tiles. In other words, the roof tile artisans responded to this stress by changing their firing methods. In contrast, other factors did not respond to the stress, and therefore constructed different transition patterns.

The evolutional mechanism of roof tiles has only been revealed by studying the evolution of each factor from multiple perspectives. In some archaeological studies, artifacts including roof tiles were classified based on limited factors that were mainly external characteristics, and that classification was regarded as a historical indicator of human behavior and culture. In previous studies of external characteristics, ancient Ryukyuan roof tiles were considered to have been influenced by some roof tile cultures of mainland Japan, the Korean peninsula, and the Chinese dynasty. However, although Ryukyuan roof tiles were transformed into unique red tiles in the Ryukyu Islands and were passed on to today's traditional culture, these cultural origins do not sufficiently explain the roof tiles' evolution. We therefore provided a new perspective

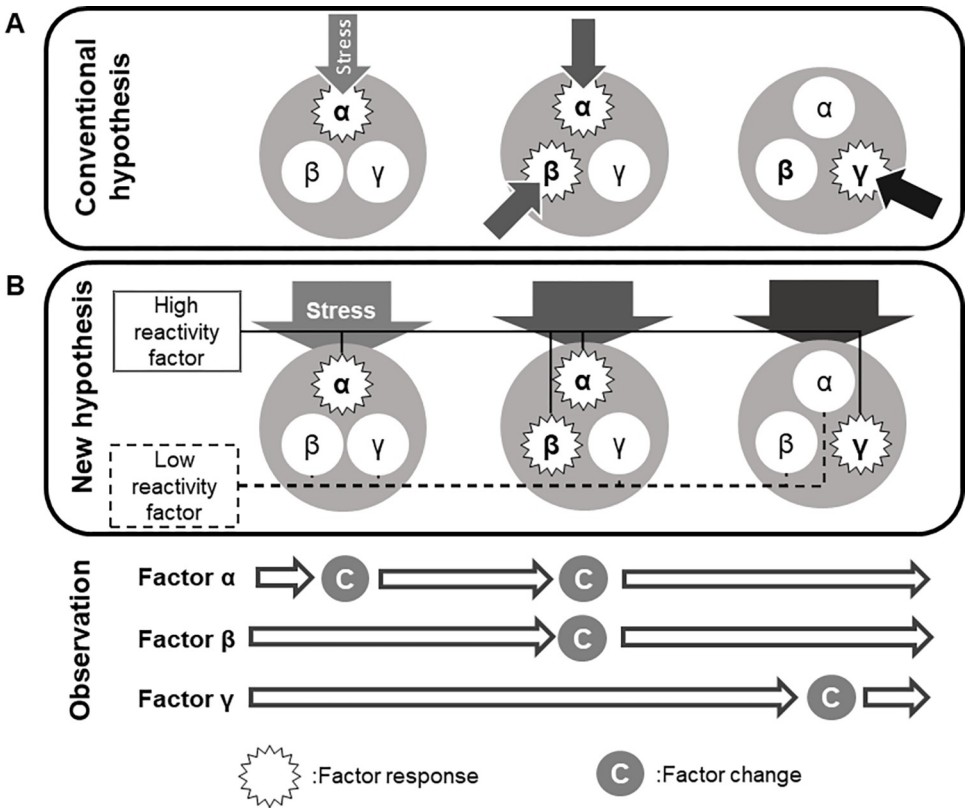

**Fig 6. Two hypotheses to explain the different transition patterns.** (A) In the conventional hypothesis, each factor is subjected to individual stresses, which results in changes to the factor. (B) In the new hypothesis, all the factors are subjected to external stresses, but each factor reacts differently, which results in changes to the factor. In both hypotheses, the observed result is the same transition pattern.

on the evolution of the Ryukyuan roof tile by proposing that it was caused by external stresses and various reactions of traits to such stressors. As a result of our multifaceted analysis, we conclude that the uniqueness of Ryukyuan roof tiles is the result of the accumulation of external stresses in Ryukyu society and the various responses and evolutions of each trait to such stressors. This is similar to the relationship between external stress and the response of each trait as well as the relationship between such traits and evolution described in the field of ecological developmental biology [39]. Many discussions of cultural evolution have applied biological evolutionary models, such as neutrality and adaptation, to the components of material culture, and our study also provides another way to think about and discuss the relationship between external stresses and responsiveness on material culture. This is a new perspective on the evolution of Ryukyuan roof tiles that could not have been achieved by discussing only the cultural origins and temporal changes based on specific factors such as appearance and form. We believe that this logic can be applied not only to Ryukyu tiles, but also to various other material cultures.

## Conclusion

This study reveals for the first time the chronological transition of internal features such as elemental composition, color tone, and internal structure of the paste of ancient Ryukyu tiles from Shuri Castle. Previous archaeological studies have explained the history of the roof tiles based on changes in their external features, but the internal features show a different transition

from these external features. It is also noteworthy that some internal features of the tiles remained largely unchanged for a very long time after the appearance of the tiles in the Ryukyu Islands. We propose that these factor-specific chronological transitions are the result of the different reactions of individual factors to external stresses based on environmental and sociopolitical contexts. In other words, the characteristics of Ryukyuan tiles did not all change at the same time, only the factors that responded to various external stresses. This shows the historical flexibility of Ryukyuan tile making, and we conclude that this represents the evolution of the roof tiles. This is a significant change from the conventional framework that views roof tile history in terms of a single transition such as typology, and it can be applied to research on various other material cultures. Our analytical and statistical methods, which were non-destructive, can also be applied to a variety of artifacts, such as pottery and ceramics, allowing us to analyze samples that include valuable materials. We believe that this will lead to new research and discussion of the evolution of material culture.

## Supporting information

**S1 Appendix. Supplementary tables S1-S11.**
(XLSX)

**S2 Appendix. Supplementary methods and figures S1-S10.**
(PDF)

## Acknowledgments

This research was funded by a grant from the *Shuri-jo* Revival Research Project of the University of the Ryukyus. This research used the Ryukyu University Museum (Fujukan) collection and samples provided by the Damaged Roof Tile Utilization Project of Okinawa Prefecture. The XGT measurements were performed at the University of the Ryukyus Center for Research Advancement and Collaboration. X-ray CT measurements were performed at the Okinawa Industrial Technology Center. We thank T. Sasaki for managing the study's resources and providing much information, advice, and help, M. Shimabukuro for help with financial operations, M. Ozaki and S. Yabu for technical assistance, T. Kawano for providing a picture of Shuri Castle, and Y. Tanahara, K. Hanashiro, and other members of the Okinawa Industrial Technology Center for providing information and advice.

## Author Contributions

**Conceptualization:** Hiroaki Aoyama, Kaishi Yamagiwa, Wataru Taira, Takeshi Kon.

**Investigation:** Hiroaki Aoyama, Kaishi Yamagiwa, Wataru Taira.

**Methodology:** Hiroaki Aoyama, Kaishi Yamagiwa, Wataru Taira.

**Project administration:** Hiroaki Aoyama.

**Supervision:** Takeshi Kon.

**Visualization:** Hiroaki Aoyama, Kaishi Yamagiwa, Wataru Taira.

**Writing – original draft:** Hiroaki Aoyama, Kaishi Yamagiwa, Wataru Taira.

**Writing – review & editing:** Takeshi Kon.

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
