## [Decision Letter · Decision Letter 0]

5 Aug 2022

PONE-D-22-12204A new perspective on the evolution of “Kawara” roof tiles in Ryukyu: a multidisciplinary non-destructive analysis of roof tile transition at Shuri Castle, Ryukyu Islands, Japan.

PLOS ONE

Dear Dr. Aoyama,

Thank you for submitting your manuscript to PLOS ONE. After careful consideration, we feel that it has merit but does not fully meet PLOS ONE’s publication criteria as it currently stands. Therefore, we invite you to submit a revised version of the manuscript that addresses the points raised during the review process.

A marked-up copy of your manuscript that highlights changes made to the original version. You should upload this as a separate file labeled 'Revised Manuscript with Track Changes'.An unmarked version of your revised paper without tracked changes. You should upload this as a separate file labeled 'Manuscript'.

We look forward to receiving your revised manuscript.

Kind regards,

Abbas Farmany

Academic Editor

PLOS ONE

Journal Requirements:

2. We note that Figure 1 in your submission contain copyrighted images. All PLOS content is published under the Creative Commons Attribution License (CC BY 4.0), which means that the manuscript, images, and Supporting Information files will be freely available online, and any third party is permitted to access, download, copy, distribute, and use these materials in any way, even commercially, with proper attribution. For more information, see our copyright guidelines: http://journals.plos.org/plosone/s/licenses-and-copyright.

Reviewers' comments:

Reviewer's Responses to Questions

**Comments to the Author**

1. Is the manuscript technically sound, and do the data support the conclusions?

Reviewer #1: Partly

2. Has the statistical analysis been performed appropriately and rigorously? 

Reviewer #1: N/A

3. Have the authors made all data underlying the findings in their manuscript fully available?

Reviewer #1: Yes

4. Is the manuscript presented in an intelligible fashion and written in standard English?

Reviewer #1: No

5. Review Comments to the Author

Reviewer #1: See uploaded PDF document for many details and feel free to share it with the authors (I wrote it assuming they would see it). In general, the analysis of the tiles themselves seems innovative, but the authors struggle, without success, to place their work in a broader social and historical context. I recommend cutting out most of this social and historical material and making what remains more precise and accurate.

6. PLOS authors have the option to publish the peer review history of their article (what does this mean?). If published, this will include your full peer review and any attached files.

Reviewer #1: No

---

## [Author Response · Author response to Decision Letter 0]

25 Sep 2022

＜Comments to Reviewer＞

Thank you for reviewing our manuscript. We think your peer review made our paper better by reaffirming the content of our discussion. On the other hand, we have only partially agreed with your suggestions. The reasons and our considerations are detailed below.

Report on “A new perspective on the evolution of “Kawara” roof tiles in Ryukyu”

This article reports the results of a sophisticated technical analysis of Shuri Castle roof tiles from different time periods and attempts to hypothesize the broader significance of the authors’ laboratory analysis. The authors also attempt to set their analysis of roof tiles in a historical context.

I am not qualified to comment on the technical analysis and methodology itself. Because that realm appears to be the authors’ area of expertise, I assume in this report that their analysis is sound. My comments deal with roof tile geography, historical context, the authors’ attempt to link their technical findings with social change/social history, and some aspects of writing. Let me say at the outset that there are problems in each of these areas. I think they can be resolved or partially resolved, but doing so will require significant effort. I have underlined some words and phrases for emphasis.

Roof Tile Geography

As the authors know, Shuri Castle was not the only large stone-walled gusuku with roof-tiled structures dating from the fourteenth century. Of course, there are pragmatic constraints with respect to the size of the sample set in a study like this one. So I can well understand why the authors analyze only tiles from Shuri Castle. However, there it least one major tile-related issue about which the authors are silent. Scholars have long speculated about the significance of roof tiles made in almost exact imitation of late Goryeo (Korean) tiles. (The authors often refer to Goryeo as Kōrai [or Kourai], the Japanese pronunciation of the Goryeo dynasty in Korean history, 918-1392). These tiles are found at Shuri Castle, Urasoe Castle, and Katsuren Castle. One issue regarding the tiles is their date. In all three cases, the tiles display the date癸酉. Some scholars argue the sexagenary characters indicate 1273 and others argue for 1333. Ikeda Yoshifumi, for example, favors 1333 partly on the grounds that large stone-walled gusuku themselves came into existence only in the fourteenth century. I find Ikeda’s argument convincing. Other scholars, especially those who tend to regard Ryukyu’s official histories as accurate sources, favor 1273 because of the claim that Eiso resided at Urasoe castle (although some scholars dispute this claim about Eiso).

In Ancient Ryukyu (University of Hawaii Press, 2013), Richard Pearson favors 1273. He also notes that analysis of the tiles indicates that the clay all came from the same quarry, located near Nago. I am away from my office as I write this report, and I do not have access to my library. So unfortunately, I cannot provide page numbers. However, Person’s discussion of roof tiles is extensive and can easily be found via the book’s index.

I am aware of over twenty different works that discuss these Korean-style roof tiles and their significance, and I am sure there are many other works I have not seen. So what is the point with respect to the current manuscript? Obviously the authors have analyzed only tiles from Shuri Castle at this point. Instead of vague speculation about social matters (more on that below) in an attempt to justify the article, perhaps a better (or additional) approach might be to discuss the possible significance of their analysis with respect to the Korean tile academic debate. Perhaps their analysis, if extended to the other two locations, can she useful light on this matter.

[One other background point, which might be worth mentioning, is that stone-walled gusuku in the Ryukyu islands [as opposed to trench-and-earthworks gusuku like Sashiki ui gusuku or Nago gusuku] predate Japanese castles by at least a century. So from where did the knowhow to build them come? Chinese castles are entirely different in terms of their construction. But Korean mountain fortresses (山城), of which there are over 2,000 extant examples, resemble Ryukyuan stone-walled gusuku almost exactly (including the three major types of wall construction). One excellent source for the Korean fortresses is Guglib munhwajae yeonguso 國立文化財硏究所 (National Research Institute of Cultural Heritage, Korea). 2012. Hangug gogohag jeonmun sajeon, seonggwag bongsu pyeon 韓國考古學專門事典: 城郭, 烽燧 篇. Daejeon, South Korea: Guglib munhwajae yeonguso.]

＜Comments to Reviewer＞

First, we focused on Shuri Castle, because it is one of the cultural centers and tiles have been used for a relatively long period of time in the same place. We approach the evolution of the tiles by scientifically analyzing the changes in the tiles limited to Shuri Castle. Therefore, as the reviewer pointed out, the analysis was conducted only for the Shuri Castle tiles. 

We are also aware that there is some debate about the date of manufacture of the Kourai tiles, as has been pointed out, but we understand that no definite decision has yet been reached at this time.

And we inferred social issues based on the data analyzed in this study, not merely vaguely imagined. In those analyses we used three types of tiles (Yamato, Kourai, and Mincho), not only Kourai. Through comparisons of these three types, we discussed the relationship between technological changes and social context.

Historical Context

Of course, none of us can be experts on everything. The literature on Ryukyuan history and archaeology is vast, and over the past fifteen years or so, our understanding of Ryukyu’s deep past has changed significantly. Obviously, certain points

are debated and many questions remain unsettled. My comments below are based on a general consensus of recent (past 15 years) work in ancient Ryukyu. Nearly all of this research is in Japanese, but one important, relevant book in English is Richard Pearson’s work mentioned above. In the paragraphs below I will paste in statements from the manuscript and comment on them (manuscript line numbers included). Overall, the problem is a lack of specificity with respect to time periods and concepts.

In the Ryukyu Islands, Kawara was introduced 72 after about the 13th or 14th century AD, when multiple types of roof tiles appeared with several origins 73 assumed to be mainland Japan, the Korean peninsula, or mainland China. The ancient roof tiles, as a 74 typical artifact, were evidence of the prosperous international relations between the Ryukyu kingdom 75 and its surrounding countries.

The authors cite an early article by Ikeda Yoshifumi. I recommend they read some of his work from 2012 onward—he often repeats himself, so any of several book chapters or articles will do. As Ikeda points out, although some gusuku sites existed in the 13th century, the stone walls and tile-roofed structures all date from the 14th century. As for origins of gusuku era kawara: mainland Japan and Korea, yes; China, no.

＜Comments to Reviewer＞

Since Ikeda noted that matter in his 2012 reference (Ikeda 2012), we found no need to change the citation. In addition, reviewer has noted that Ikeda has made it clear in his recent writings that tiled buildings appeared after the 14th century, but this is not accurate. Ikeda (2019), for example, states that "the appearance of buildings with stone walls and platforms is in the fourteenth-century period" (Ikeda 2019:29). However, it is not clear whether the only buildings with platforms were tile-wiping buildings, and Ikeda does not state that tile-roofed buildings appeared after the 14th century. There are several possible dates for the appearance of Kourai roof tiles including the late 13th century, and the issue has not been settled. It is not appropriate to define the appearance of the roof tiles as being after the 14th century on the basis of the appearance of the platform building alone, and we have used a wide range of dates in our manuscript. Furthermore, we do not discuss in this section the date of appearance of buildings shingled with Kourai style roof tiles. Therefore, we have not adopted the reviewer's recommended citation.

[Reference]

Ikeda, Y. 2019. Ryukyu Rettou-shi wo Horiokosu: Juuichi-Juuyonseiki no Ijuu, Koueki to Syakai-teki Henyou (The Unearthed History of Ryukyu Islands: Maritime Trade, Migration, and Social Transition). Chuusei gaku kenkyu-kai (ed.). Ryukyu no Chuusei (Middle Age of Ryukyu). Koushi Shoin inc, Tokyo: 13-37.

Now, for something more subtle but very important. It is common for many writers to throw around the term “Ryukyu kingdom” casually and loosely. This term is very tricky because it means two different things across time. The first meaning (1370s-ca. 1500): some people (or fictitious people) on the island of Okinawa conducted official trade with China starting in the 1370s. These people held the title “wang” which is always translated as “king.” However, the title wang was simply a license to conduct official trade with China. It says nothing about the domestic circumstances of these “kings,” of which there were as many as four at a time in Okinawa, ca. the late 1300s. The second meaning (ca. 1500-1879). When did a Ryukyu kingdom in the usual sense of the term (a centralized government headed by a monarch) come into existence? The traditional date is 1429, but that is clearly a creation of later centuries. There is no exact date, but one possibility is Shō Shin’s successful (for him) invasion of Yaeyama in 1500. Nevertheless, additional warfare was needed to complete his conquests of the Ryukyu islands. Moreover, the earliest extant written documents used for domestic governance (jiresisho) are from the late 1520s. There is much more to say, but I would put the start date of a genuine Ryukyu kingdom at about 1520 or 1530—roughly 100 years later than the usual claim based on the official histories.

So “the prosperous international relations between the Ryukyu kingdom and its surrounding countries” doesn’t really make sense with respect to the 13th or 14th centuries. There was no Ryukyu kingdom then, and as I mentioned, the first people holding the title “king” (meaning trade license) appeared in the 1370s.

＜Comments to Reviewer＞

Pearson (2013) described the beginning of the Ryukyu Kingdom period to AD 1429 (p. 234). The Okinawa Archaeological Society (2019) also described the beginning of the Ryukyu Kingdom period in the early 15th century, including AD 1429 (some argue the late 14th century); the theory of the late 16th century is not a consensus within Okinawan archaeologists, and we do not adopt it.

[References]

Pearson, R. 2013. Ancient Ryukyu: An Archaeological Study of Island Communities. University of Hawai'I Press, Honolulu.

Okinawan Archaeological Society (ed.). 2018. Nantou Kouko Nyuumon: Horidasareta Okinawa no Rekishi, Bunka (Japanese Southern Islands Archaeology: Unearthed History and Culture of Okinawa). Borderink, Okinawa.

Here is a more accurate statement: Kawara first appeared in Okinawa in the early fourteenth century in connection with the construction of large fortresses (gusuku). The Korean peninsula was one source of kawara or kawara-making knowhow, and it is also likely that mainland Japanese kawara or kawara-making knowhow also came into the Ryukyu islands at about this time. The arrival of kawara corresponds to a time when local powers based in harbors in Okinawa pursued vigorous regional trade throughout the East China Sea.

<For a little more context, these local powers in Okinawa, based at places like Katsuren, Nakagusuku, Itokazu, Sashiki, Urasoe, and Nakijin, although often called aji today, were wakō 倭寇. The official trade with China that began in the 1370s was, above all else, a wakō control policy on the part of the Ming dynasty. Inamura Kenpu 稲村賢敷published a book on this topic in 1957 (琉球諸島における倭寇史跡の研究), and more recently, scholars such as Tanigawa Ken’ichi and Yoshinari Naoki have written about wakō and Ryukyu extensively. The Ryukyu islands of the 1300s and 1400s were not part of an orderly kingdom. They were a frontier region.>

79 Ryukyuan roof tiles appeared at least as early as the 13th to 14th centuries AD (This period corresponds 80 to the Gusuk period in 10th to 14th century of Ryukyu Islands) and are presumed to have been used in 81 Gusuku, the large stone-walled castles and fortresses that are representative of this period. The oldest 82 tiles are classified into two styles: Kourai and Yamato.

Again, instead of “as early as the 13th to 14th centuries” I recommend “circa the early 14th century.” Roof tiles were not “presumed to have been used” in gusuku. Instead, they were used in gusuku, specifically for the roofs of certain structures inside these gusuku. This second point is mainly an issue of diction or style, and it is subtle. For academic writing in English, I recommend clear declarations when the evidence supports them (more on this topic below).

＜Comments to Reviewer＞

As we have written before, there are various theories on the date of appearance of tiles in Ryukyu, and we cannot make a definite determination at this time. Therefore, we have given a wide range of date, "as early as the 13th to 14th centuries"

The Kourai style is associated with Korea’s 83 Goryeo Kingdom (AD 918–1392) due to external characteristics, such as design, shape, paddling traces, 84 imprinting, and inscriptions on their surface [4–6]; specifically, they are assumed to be connected to 85 refugees from the failure of the “Sambyeolcho” Rebellion in Goryeo (AD 1273)

This is a good point. Indeed, as Kurima Yasuo and others have argued, all major changes in early Ryukyuan history came from influxes of people from outside the islands. For example, during the early gusuku era (11th-12th centuries; I would date whole era as the 11th-15th centuries), migrants from Japan (mainly) and Korea swamped out the sparse indigenous population of Jōmon people (confirmed abundantly by recent ancient DNA studies). Moreover not only did the Sambyeolcho Rebellion have an impact on Okinawa, the gradual collapse of the Goryeo dynasty from ca. 1350 onward did as well. For example, the occupants of Urasoe Castle, operating under the name Satto, almost certainly came from Korea.

In the 14th and 15th centuries AD, the Ryukyu archipelago was unified into the Ryukyu Kingdom 92 and became part of the Chinese dynasty’s tributary system.

Because there is so much confusion about this era, please be more specific and accurate. Here is one possibility: From the 1370s, local powers in Okinawa engaged in official trade with China.

That is all I would say. The “Ryukyu archipelago” did not become unified under Shuri’s control until the 16th century. I realize the authors are not historians, but brief consideration of well-documented events such as Shuri’s invasion of Yaeyama in 1500, Shuri’s conquest of Kumejima in 1506, and Shuri’s conquest of Yonaguni (ether the 15-teens or 1520s) indicate the approximate time when all of the Ryukyu islands became, at least nominally, under Shuri’s control: the early 16th century.

＜Comments to Reviewer＞

We changed to “the Okinawa island” as suggested (p5, line 92).

Although the 125 Castle’s origins remain unclear, some historical and archaeological research has suggested that the main 126 palace was constructed in the mid-15th century (“Ankoku-zan Juka-boku no Kihi”: inscription on stone 127 monument built in 1427). It is presumed to have had a tile roof at this time because of the large number 128 of tiles excavated [12–14]. This main palace was destroyed by fire at least three times (AD 1453, 1660, 129 1709) (“Kyuyou”, a historical record written in AD 1743–1745).

The first sentence is odd. Although indeed we do not know the exact month or year that Shuri Came into being, 1405 or 1406 is very close. It is also important to note that the current, Chinese-looking version of the castle dates from about the 1490s or ca. 1500-1510 (part of the larger process of the kingdom coming into being). We know this in part because of the detailed descriptions of Shuri Castle by Koreans temporarily residing in Okinawa in the mid 1400s. They describe a castle that looked much different from its later iterations. These Korean descriptions of Shuri castle are widely discussed in the Japanese literature. Takara Kurayoshi, for example, includes a summary in many of his recent books and book chapters.

＜Comments to Reviewer＞

We do not agree with the reviewer's consideration because there are no citeable references. Therefore, we left it unchanged.

Because no clear traces of firing or production facilities for 13th- to 15th-century roof tiles have 391 been found in the Ryukyu Islands, it has not been possible to conclude whether they were produced in 392 this region or were imported. However, our results suggest that, at the very least, the technical 393 background of the production of these two roof tile styles at Shuri Castle (perhaps also at Urasoe Castle) 394 had similarities that can be detected in their color, paste characteristics, and internal structure. This might 395 mean that both the Kourai and Yamato roof tiles were made in the Ryukyus.

I agree with the final sentence, and indeed I would state the conclusion with greater strength. The first statement might be correct, although furnaces and kilns using Korean technology existed Tokunoshima from the late 11th century (to produce Kamuiyaki) and temple bells were cast in Shuri castle (and probably other locations) during the 15th century. So the basic knowhow probably existed during the 14th and 15th centuries. In any case, I think that the origin of the clay for the Korean-style roof tiles in Shuri castle, Urasoe Castle, and Katsuren has been known definitely for some time—an area near Nago, whose precise name escapes my mind. Again, my apologies for not providing specific page numbers, but see Pearson’s discussion for the details and relevant references in Japanese. So I suggest re-writing this paragraph to make the point more forcefully. The overall situation was technical knowhow (and almost certainly actual human technicians), which came from Korea and mainland Japan, but production taking place in Okinawa.

＜Comments to Reviewer＞

Our data did not reveal the origin if the technologies, nor did we identify its provenance. Therefore, we have taken the current expression and cannot say much more than that.

Although Ryukyu had the opportunity to import 461 techniques from Japan and China, it did not use these techniques but developed its own unglazed reddish 462 roof tiles. The reasons for this remain unclear, but as Ryukyu was in the complicated situation of being 463 under the control of Satsuma while having a feudal relationship with China [36], it is possible that they 464 did not prioritize the incorporation of a single country’s technology. The reddish color may also have 465 been a result of the feng-shui values that affected the policies of the Ryukyu Kingdom from the Qing 466 dynasty at that time [37]. This suggests that the background to the appearance of colored roof tiles in 467 Ryukyu encompasses not only the production technology and fuel resources, but also the socio-political 468 background. 469

The first sentence above is odd insofar as the original techniques and technology were clearly Korean (and possibly mainland Japanese) imports. But the authors transformed these outside technologies into something “uniquely” Ryukyuan in preceding paragraphs (a move probably best avoided).

More important: This whole paragraph is nothing but idle speculation by writers who clearly have only a superficial understanding of Okinawan history and society at this time. Again, I acknowledge that we cannot be experts in all fields, so I hate to sound critical. Nevertheless, a statement like “as Ryukyu was in the complicated situation of being 463 under the control of Satsuma while having a feudal relationship with China [36], it is possible that they 464 did not prioritize the incorporation of a single country’s technology” does not even make superficial sense—in addition to being nothing but speculation. Even general survey histories of Okinawa would note that during the early-modern era (ca. 1609-1879), the Ryukyu kingdom adopted extensive technologies from both China and Japan in fields like medicine, manufacturing, agriculture, and astronomy to name a few. Why, therefore, would it be reluctant to do so regarding roof tiles? And as for “The reddish color may also have 465 been a result of the feng-shui values that affected the policies of the Ryukyu Kingdom from the Qing 466 dynasty at that time,” please explain what you mean, specifically, with respect to fengshui. In any case, here and throughout all of the concluding sections, there is too much idle, unconvincing speculation.

＜Comments to Reviewer＞

We have removed the description of Feng Shui, as indicated (p 24, line 464).

However, sine this section discussed the color change in “Mincho style tiles”, the reviewer's suggestion “the original techniques and technology were clearly Korean imports” is not appropriate. We also discussed the relationship between the technology of neighboring countries and color change in Ryukyus, and only mentioned social context as one of possible explanation.

Important: the material from lines 493 to 520 is especially confusing. It all needs to be re-written. First, clarify exactly who has proposed which hypothesis. Second, explain each hypothesis better so that it is clear what the difference between them—and yours is. I have read these sentences three times and I cannot tell what exactly the authors mean.

＜Comments to Reviewer＞

In this paragraph, we discussed the transition (evolution) of tiles from a material culture perspective, based on the results obtained from the analysis of this study. In those discussions, we have also crearly presented past studies (by Dunnell, Boyd R, Richerson PJ) and then have proposed our new hypothesis. In other words, we discussed the history of Ryukyu (historical context), but rather presented a hypothesis regarding the evolution of material culture.

Also important: The authors periodically speculate about fuel shortages. For example:

In the case of transition pattern 2, it is considered 515 that the entire roof tile culture was subjected to the social stress caused by fuel shortages, but some of 516 the firing methods showed an especially strong reaction to it, which led to a transition in the color and 517 pore size of the tiles. In other words, the roof tile artisans responded to this stress by changing their 518 firing methods.

However, nowhere in the paper do the authors clearly present evidence of fuel shortages at certain times. Because this alleged fuel shortage is the only specific social factor that the authors claim (as opposed to vague speculation), it should be well explained. In other words, is there scholarly evidence for a significant fuel shortage during the early-modern era? If so present it (I cannot help here, because I am not aware of any such a claim in the literature). Or, if the physical analysis of the tiles either proves or suggests a fuel shortage, the authors need to explain the causal chain with clarity. Different firing methods? OK, that seems more reasonable and fairly obvious based on the evidence, but why does a different firing method mean that there was a fuel shortage? Perhaps it does, but if so, the authors need to explain it clearly.

＜Comments to Reviewer＞

The evidence on fuel shortages and their social context has already been clearly presented (line 452-457).

We 528 therefore provided a new perspective on the evolution of the Ryukyuan roof tile by proposing that it was 529 caused by external stresses and various reactions of traits to such stressors. As a result of our 530 multifaceted analysis, we conclude that the uniqueness of Ryukyuan roof tiles is the result of the 531 accumulation of external stresses in Ryukyu society and the various responses and evolutions of each 532 trait to such stressors.

The authors repeat some variation of this conclusion several times (indeed, too many times). It makes no sense. What exactly are “external stresses and various reactions of traits to such stressors”? Other than the alleged fuel shortage, the authors are at pains even to speculate about specific possibilities. Herein lies the paper’s major problem. The authors have conducted what appears to be an innovative analysis of roof tiles. Excellent. But they cast about, without success, trying to find some kind of larger significance for their work. Granted, coming up with something is difficult. I can think of only one thing.

＜Comments to Reviewer＞

Again, we have discussed the evolution of Ryukyuan tile from material culture perspective. The purpose of this paper is not to complement the history or origin of roof tiles in Ryukyu.

I suggest removing most or all of this material. A better approach, albeit a more modest one, might be to study the literature on the Korean roof tiles I mentioned at the start of this report and suggest how the authors’ method of tile analysis might shed useful light on this ongoing academic debate.

If indeed there are any relevant social “stressors” (a term I suggest the authors abandon) for which there is good evidence, then the authors should state their case with clarity and specificity, backed up with suitable citations to relevant scholarship.

＜Comments to Reviewer＞

" Stress" is a "cultural evolutionary” term already presented in Boyd and Richerson. This is also made clear in the manuscript (p. 26, line 498)

Writing

The quality of the writing needs improvement. Many sentences contain vague wording, and the authors use the passive voice excessively (e.g., “is presumed” “can be assumed” etc.). Organization at the level of paragraphs tends to break down in the latter third of the paper. Small spelling errors are common. And there is more. I am not going to go on at length about writing, leaving that matter to the editor. However there is one important issue, the spelling or Romanizing of Japanese words.

The authors spell Japanese words inconsistently. I provide some typical examples:

<(“Kyuyou”, a historical record written in AD 1743–1745).> The typical manner of spelling this word (球陽) would be Kyūyō. Note that both the “u” vowel sound and the “o” vowel sound are long. The conventional way to indicate long vowels is with macrons, ū and ō in this case. Writing out the vowels, while not conventional, does make sense and indeed it corresponds to the Japanese kana spelling きゅうよう. However, in this case, the Roman letter spelling should be Kyuuyou, not Kyuyou.

Looking at the bibliography, the Romanization of Japanese titles is inconsistent in numerous ways. The sound fu ふ is often spelled fu, but sometimes it appears as hu (as in huusui 風水). These sounds should all appear spelled as “fu” or “fuu.” Moreover, long vowels are indicated in some cases but not others, and capitalization is irregular. To take but one example, here is a book title:

Shuen to chushin no gainen de yomitoku higashi ajia no Etsu Kan Ryuu: Rekishigaku kokogaku kenkyu kara no shiza

Using the authors’ system it should appear as (note also that Higashi Ajia [East Asia], is a proper noun):

Shuuen to chuushin no gainen de yomitoku Higashi Ajia no Etsu, Kan, Ryuu: Rekishigaku, koukogaku kenkyuu kara no shiza

Or more conventionally:

Shūen to chūshin no gainen de yomitoku Higashi Ajia no Etsu, Kan, Ryū: Rekishigaku, kōkogaku kenkyū kara no shiza

Almost every entry in the bibliography contains Romanization errors, as do some words in the main text.

＜Comments to Reviewer＞

This paper has been reviewed in English by a professional agency. Variants in romanization were the result of following the original notation of the journals cited.

---

## [Decision Letter · Decision Letter 1]

31 Oct 2022

A new perspective on the evolution of “Kawara” roof tiles in Ryukyu: a multidisciplinary non-destructive analysis of roof tile transition at Shuri Castle, Ryukyu Islands, Japan.

PONE-D-22-12204R1

Dear Dr. Aoyama,

We’re pleased to inform you that your manuscript has been judged scientifically suitable for publication and will be formally accepted for publication once it meets all outstanding technical requirements.

Kind regards,

Abbas Farmany

Academic Editor

PLOS ONE

Additional Editor Comments (optional):

Reviewers' comments:

Reviewer's Responses to Questions

**Comments to the Author**

1. If the authors have adequately addressed your comments raised in a previous round of review and you feel that this manuscript is now acceptable for publication, you may indicate that here to bypass the “Comments to the Author” section, enter your conflict of interest statement in the “Confidential to Editor” section, and submit your "Accept" recommendation.

Reviewer #1: (No Response)

2. Is the manuscript technically sound, and do the data support the conclusions?

Reviewer #1: Yes

3. Has the statistical analysis been performed appropriately and rigorously? 

Reviewer #1: N/A

4. Have the authors made all data underlying the findings in their manuscript fully available?

Reviewer #1: Yes

5. Is the manuscript presented in an intelligible fashion and written in standard English?

Reviewer #1: Yes

6. Review Comments to the Author

Reviewer #1: In a few cases the authors correctly note that my comments missed or misconstrued something (e.g., regarding the term "stress"). Overall, however, the authors were dismissive regarding many of my substantive comments. I have no intention of spending any more of it on this matter except for one item:

It is also important to note that the current, Chinese-looking

version of the castle dates from about the 1490s or ca. 1500-1510 (part of the larger process of the

kingdom coming into being). We know this in part because of the detailed descriptions of Shuri Castle

by Koreans temporarily residing in Okinawa in the mid 1400s. They describe a castle that looked much

different from its later iterations. These Korean descriptions of Shuri castle are widely discussed in

the Japanese literature. Takara Kurayoshi, for example, includes a summary in many of his recent

books and book chapters.

＜Comments to Reviewer＞

We do not agree with the reviewer's consideration because there are no citeable references. Therefore,

we left it unchanged

Here are your references:

Higashionna Kanjun _Higashionna Kanjun zenshuu_, vol 3, pp. 81-85

Joseon Royal Records (朝鮮王朝実録), Entry nos. 125 & 126 (1462) in the edition edited by Ikeya Machiko, Uchida Akiko, and Takase Kyouko, _Chousen ouchou jitsuroku Ryuukyuu shiryou shuusei_. (The account is by Ryang Seong).

Takara Kurayoshi, _Ryuukyuu oukokushi no tankyuu_ (Ginowan, Youju shorin 2011), p. 32.

Makishi Youko, "Ryuukyuu oukokujuugoseiki chuuki ikou no kinaisei-tekina tokuchou to oujou girei no hensen," In _Okinawa bunka kenkyuu_, No. 38 (2012), p. 162.

--end--

7. PLOS authors have the option to publish the peer review history of their article (what does this mean?). If published, this will include your full peer review and any attached files.

Reviewer #1: No

---

## [Editor Report · Acceptance letter]

7 Nov 2022

PONE-D-22-12204R1 

A new perspective on the evolution of “Kawara” roof tiles in Ryukyu: a multidisciplinary non-destructive analysis of roof tile transition at Shuri Castle, Ryukyu Islands, Japan. 

Dear Dr. Aoyama:

I'm pleased to inform you that your manuscript has been deemed suitable for publication in PLOS ONE. Congratulations! Your manuscript is now with our production department. 

Kind regards, 

on behalf of

Dr. Abbas Farmany 

Academic Editor

PLOS ONE